

# Shifts in water column microbial composition associated to lakes with different trophic conditions: "Lagunas de Montebello" National Park, Chiapas, México

Alfredo Yanez-Montalvo[1,2,*], Bernardo Aguila[1,3,*], Elizabeth Selene Gómez-Acata[1], Miriam Guerrero-Jacinto[1,4], Luis A. Oseguera[5], Luisa I. Falcón[1] and Javier Alcocer[5]

[1] Instituto de Ecología, Universidad Nacional Autónoma de México, Mérida, YUCATÁN, Mexico
[2] Unidad Chetumal, El Colegio de la Frontera Sur, Chetumal, QR, Yucatán, Mexico
[3] Posgrado en Ciencias Biológicas, Universidad Nacional Autónoma de México, Coyoacán, CdMx, Mexico
[4] Posgrado en Ciencias del Mar y Limnología, Universidad Nacional Autónoma de México, Merida, Yucatan, Mexico
[5] Grupo de Investigación en Limnología Tropical, FES Iztacala, Universidad Nacional Autonoma de México, Iztacala, Estado de México, Mexico
[*] These authors contributed equally to this work.

Corresponding authors
Luisa I. Falcón,
falcon@ecologia.unam.mx,
luisaifalcon@gmail.com
Javier Alcocer, jalcocer@unam.mx

## ABSTRACT

Eutrophication is a global problem causing the reduction of water quality and the loss of ecosystem goods and services. The lakes of the "Lagunas de Montebello" National Park (LMNP), Chiapas, Mexico, not only represent unique and beautiful natural scenic sites in southern Mexico but are also a national protected area and RAMSAR site. Unfortunately, some of these lakes started showing eutrophication signs since 2003. Anthropogenic activities (*e.g.*, land-use change from forested to agricultural and urban development) are leading to water quality and trophic state alterations of the lakes of the LMNP. This study shows the results of a coupled limnological characterization and high-throughput sequencing of the V4 hypervariable region of the 16S rRNA gene to analyze the microbial composition of the water column in a set of oligotrophic and eutrophic lakes. Chlorophyll a (Chl-a) was the main environmental parameter correlated with the trophic conditions of the lakes. Although the microbial diversity was similar, the microbial composition changed significantly from oligo to eutrophic lakes. Proteobacteria, Firmicutes, and Cyanobacteria were the main components of oligotrophic lakes, and Cyanobacteria, Proteobacteria, and Bacteroidetes of eutrophic lakes. While Acinetobacter (Proteobacteria) and *Cyanobium* (a unicellular cyanobacterium) dominated in oligotrophic lakes, the filamentous, bloom-forming, and toxin-producing cyanobacteria *Planktothrix* was the dominant genus in eutrophic lakes. High-throughput sequencing allowed the detection of changes in the composition of the microbial component in oligotrophic lakes, suggesting a shift towards eutrophication, highlighting the relevance of sensitive monitoring protocols of these ecosystems to implement remediation programs for eutrophicated lakes and conservation strategies for those yet pristine.

## INTRODUCTION

Microorganisms play a predominant role in aquatic ecosystems, as they are primary producers, regulate nutrient availability, degrade organic matter, and move or store elements through biogeochemical cycles (*Falkowski, Fenchel & Delong, 2008*; *Pernthaler, 2013*). Understanding microbial diversity and composition is crucial for assessing the ecosystem health of freshwater lakes by relating changes to natural-temporal events, disturbance processes, and anthropogenic perturbations (*Jiao et al., 2018*; *Zhu et al., 2019*).

Freshwater environments have been less studied than marine (*Humbert et al., 2009*), yet they are essential for human activities and provide a wide range of ecosystem goods and services (*Ávila et al., 2017*; *Dodds & Whiles, 2020*). However, eutrophication of lakes represents a major problem associated with poor water quality, economic losses, and diseases (*Du et al., 2019*; *Khan & Ansari, 2005*; *Smith & Schindler, 2009*). Eutrophication is a widespread phenomenon worldwide; more than 40% of lakes and reservoirs in the Americas are eutrophic, which poses a huge problem in developed and developing countries (*Ansari & Gill, 2014*; *Carpenter, 2005*; *Vander Gucht et al., 2005*). The most significant factors in eutrophication are related to an increase in nutrients, especially phosphorus (P) and nitrogen (N) (*Harper, 1992*; *Xu, Thornton & Post, 2013*), which are commonly released by wastewater discharge, agriculture, and other anthropogenic activities that override natural processes (*Lake et al., 2001*; *Moss et al., 2013*). During eutrophication, both abiotic and biotic variables change rapidly, affecting the system's stability and modifying the ecosystem's taxonomic and functional diversity (*Jeppesen et al., 2005*; *Zhou et al., 2017*).

The "Lagunas de Montebello" National Park (LMNP), Chiapas, Mexico, is one of the most beautiful scenic sites in southern Mexico, known for its crystal-clear oligotrophic lakes, which are surrounded by tropical rainforests. It was declared a protected natural area in 1959 and acknowledged as Ramsar Site 1325 in 2003 (*Gonzalez-del Castillo, 2003*). In addition to its relevance as a recreational/touristic destination, these lakes are also used for water supply, fisheries, agriculture, and cattle raising, among others. The LMNP constitutes an area of great economic, social, and political interest. Agriculture has been the most important economic activity in the region; *e.g.*, from 1992 to 2014, cultivated areas increased (13.4%), and forested areas decreased (67%) within the protected area (Fig. 1, Table S1) (*Reyes-Ramos, 1992*; *Zárate-Toledom, 2015*). To date, there are evident signs of lake eutrophication (*e.g.*, green color, algal scums, fetid odor) (*Alcocer et al., 2021*; *Fernández, Alcocer & Oseguera, 2021*). The NW section of the LMNP, a plateau area, holds a group of primarily eutrophic lakes, while lakes in the SE mountain region are oligotrophic (*Alcocer et al., 2018*). Eutrophic lakes have decreased their capacity to provide ecosystem goods and services (*Ávila García et al., 2020*; *Fernández, Alcocer & Oseguera, 2021*). Also, there is a potential health risk for water usage in the region since eutrophic lakes have been reported to exhibit cyanobacterial blooms which produce cyanotoxins (*Fernández, Alcocer*

*& Oseguera, 2021*). The situation of the eutrophication of the LMNP and its social and economic context is complex and has raised considerable concerns locally and regionally.

Nowadays, the LMNP constitutes a natural laboratory for understanding the shifts in the microbial composition of otherwise similar water bodies (*e.g.*, same climate, origin, age, geological setting) but with different trophic conditions (*i.e.*, oligotrophic and eutrophic). Therefore, this study aimed to characterize the bacterial and archaeal components of the water column of lakes with different trophic conditions along with their environmental variables. The main goal was to understand how eutrophication's microbial (bacterial and archaeal) composition has been affected. For this purpose, we addressed the following research questions: (1) How have the main water variables changed from oligo- to eutrophic conditions? (2) How has the microbial composition changed from oligo- to eutrophic conditions? Knowledge of the shifts in the composition of the microbial component of the water column will help identify lakes in the early stages of trophic change and allow for prompt interventions to reverse and control eutrophication. Our central hypothesis was that land-use changes in the surroundings of the lakes and other anthropogenic regional activities (*e.g.*, urban and industrial development) had affected the trophic conditions of the lakes along with their microbial components. Specifically, we expected these anthropogenic activities to cause (1) reduced water quality (higher turbidity) and higher trophic conditions (higher chlorophyll-a -Chl-a- concentrations); (2) a shift in the microbial composition; and (3) a reduction in the microbial diversity in the eutrophic compared to oligotrophic lakes. To answer the research questions and test the proposed hypothesis, our approach was to evaluate the changes in (a) water quality and trophic conditions of the lakes using data derived from vertical profiles of environmental variables and Chl-a concentration along the water column, as well as (b) microbial composition of the lakes using data derived from discrete samples along the water column.

## METHODOLOGY

### Study area

Located in the southeastern Mexican state of Chiapas, the LMNP extends almost 65 km$^2$ (16°04″–16°10′16°10′N, 91°37″–91°47′4091°47′40W, 1,200–1,800 m a.s.l.). Hydrologically it belongs to the RH 30 Grijalva-Usumacinta region. Lower Cretaceous limestone covers this endorheic basin that extends into Guatemala, holding a karstic lake district. The Río Grande de Comitán river flows from the NW to the SE along the primary tectonic alignments and fracture network (*Durán Calderón et al., 2014*). Climate is temperate with mean annual values of 17.3 °C, precipitation of 2,279 mm, and evaporation of 948 mm (*García, 2004*).

The LMNP was decreed a National Protected Area in 1959 and declared a Ramsar Site (number 1325) in 2003 (*Alcocer et al., 2016*; *CONANP-SEMARNAT, 2007*). The lakes of the LMNP are aligned with a geologic fracture network where plateau (NW) and mountainous (SE) regions are recognized. Twelve lakes were chosen (Fig. 2; Tables S2 and S3) to embody the heterogeneity of the lake district, this is, lakes from the plateau to the mountainous regions, with different morphometric features (extent and depth) and trophic conditions

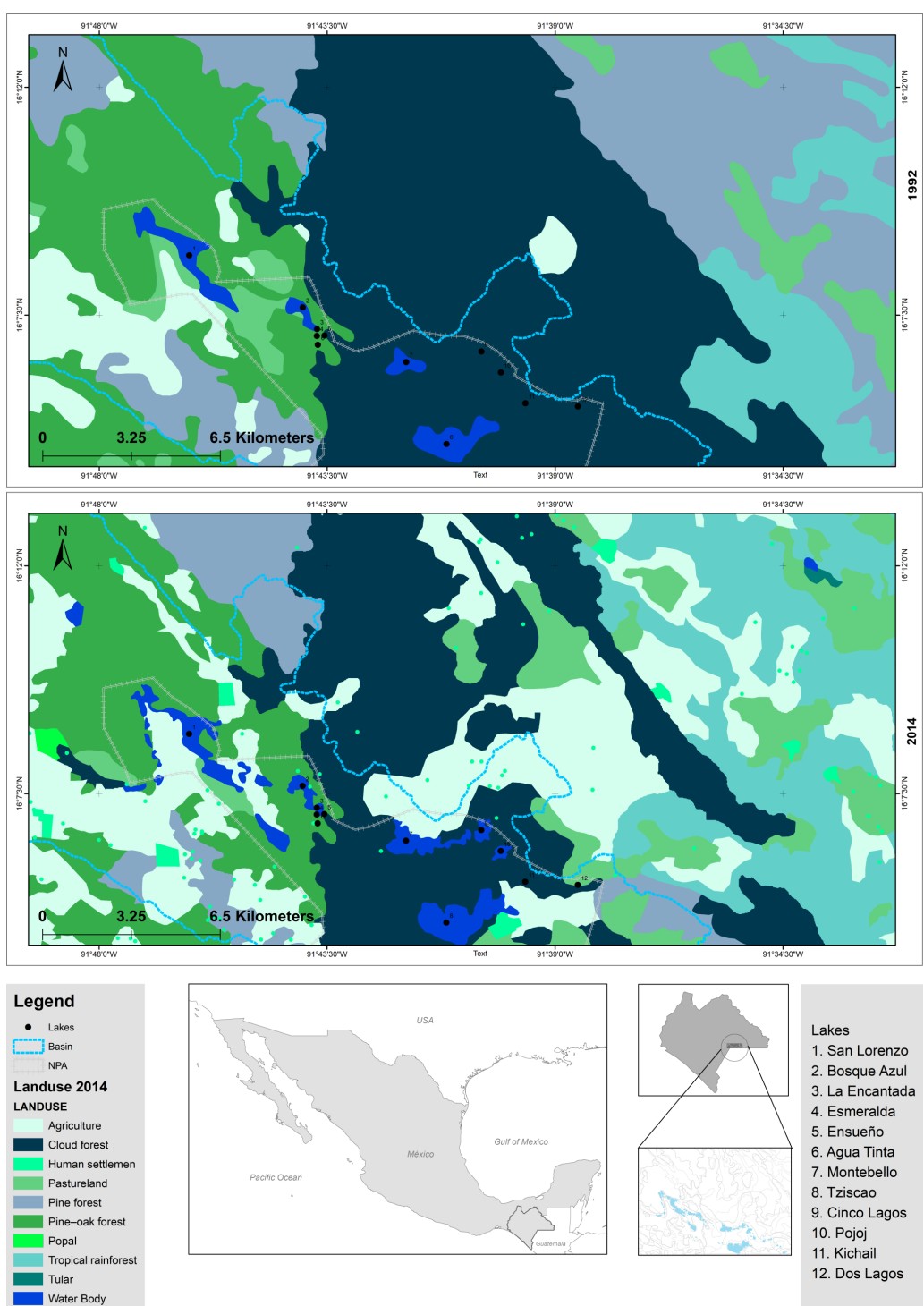

**Figure 1** **Comparison of modifications in land use of LMNP from 1992 to 2014.** Data collected from INEGI (https://www.inegi.org.mx/).

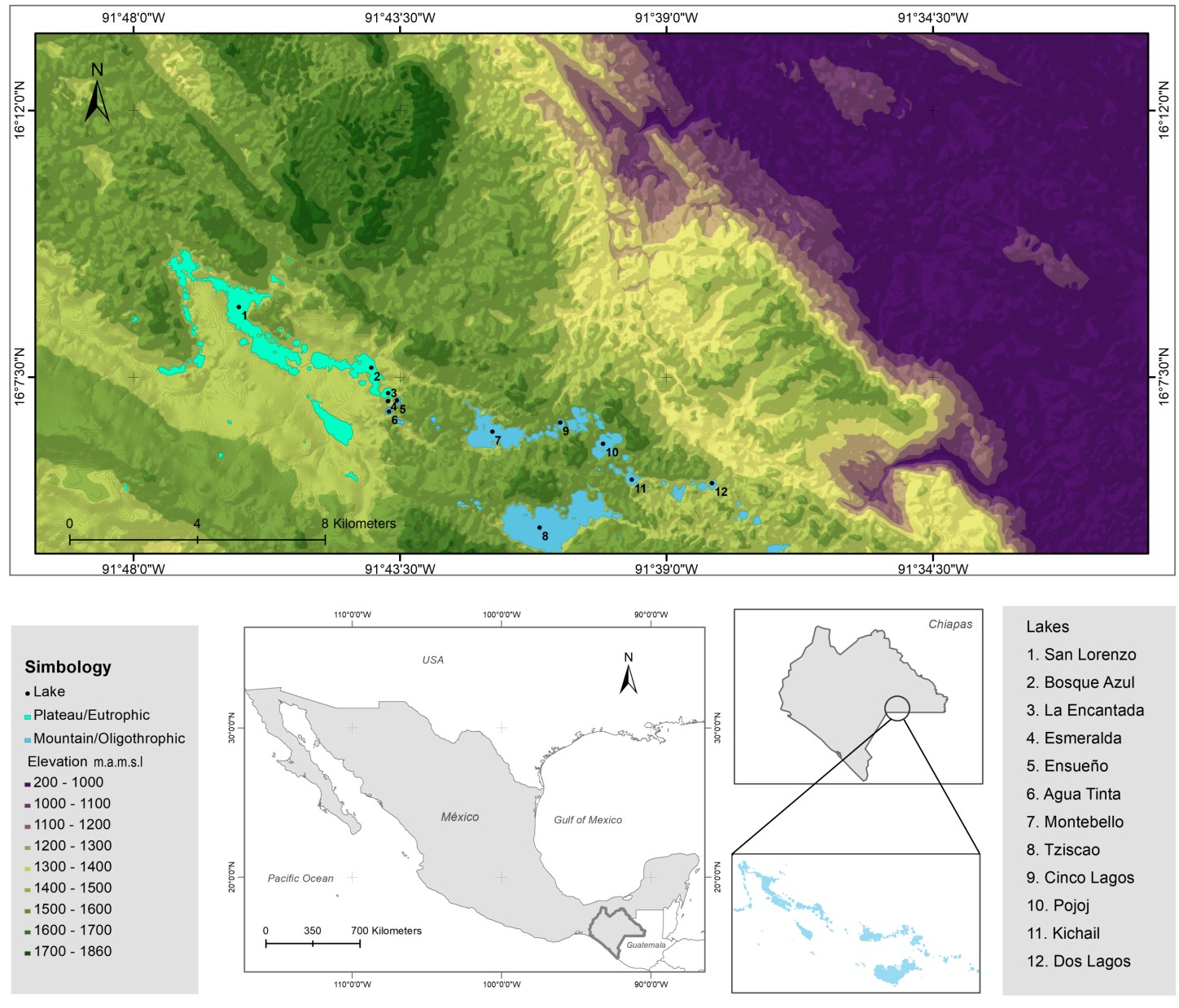

**Figure 2** **Topographic map of the LMNP.** Sampled lakes are indicated with black dots. Plateau/eutrophic lakes in green, mountain/oligotrophic lakes in blue. Data collected from INEGI.

(from oligo to eutrophic). There are three plateau/eutrophic lakes (San Lorenzo, Bosque Azul and La Encantada) and nine mountain/oligotrophic lakes (Esmeralda, Agua Tinta, Ensueño, Montebello, Tziscao, Cinco Lagos, Pojoj, Dos Lagos and Kichail). Esmeralda is a "functionally shallow" (warm polymictic) lake, while the other 11 lakes are deep, warm, and monomictic (*Cortés-Guzmán, Alcocer & Oseguera, 2019*). The trophic classification was designated according to *Vargas-Sánchez, Alcocer & Oseguera (2022)*.

## Physical and chemical characterization, water sampling, and chemical analyses

The sampling campaign was carried out at the end of March 2017. *In situ* vertical profiles of temperature, dissolved oxygen, pH, and electrical conductivity ($K_{25}$) were measured with a Hydrolab DS5 multiparameter water quality probe (1m vertical resolution). Photosynthetic active radiation (PAR = 400–700 nm) was measured with a Biospherical PNF-300 profiling natural fluorescence probe (0.25 m vertical resolution) at the central and deepest sections of the lakes except in Tziscao, where three sites were recorded due to its extent. Secchi disk depth ($Z_{SD}$) was recorded at each sampling site, and the euphotic zone ($Z_{EU}$ = 1% surface PAR) was calculated based on PAR profiles.

Water samples for Chl-a concentration and microbiological analyses were collected at different depths to better represent the water column heterogeneity according to *in situ* recorded environmental profiles (Table S3). Water samples for Chl-a concentration analysis were filtered through Whatman 0.7 $\mu$m (GF/F) membranes. After filtration, pigments were extracted from the filters with 90% acetone after incubation at 4 °C overnight. Samples were analyzed with a Turner Designs TD 10 AU fluorometer (EPA method 445.0) (*Arar & Collins, 1997*).

## Statistical analysis

The lakes of Montebello were classified according to their physical and chemical characteristics using cluster analysis (CA) by the union between groups and the square of the Euclidean distance. To determine the most critical variables in the classification, a principal component analysis (PCA) was performed. The physical and chemical variables used in CA and PCA were transformed with Z to obtain a normal distribution and to maintain the values on an acceptable scale avoiding a bias in the results. Multivariate analysis and data transformation were done with SPSS v24 software (IBM, Armonk, NY, USA).

## Microbial composition and diversity

Water samples for molecular characterization of the microbial component (Bacteria + Archaea) were collected (Table S3) in 2 L sterile bottles and kept at 4 °C during transport to the laboratory, where they were filtered through 0.22 $\mu$m pore size membranes (Durapore, Millipore) which were stored in sterile 1.5 ml containers in liquid nitrogen. DNA was extracted using the DNeasy PowerWater® Kit, following the manufacturer's instructions.

The V4 hypervariable region of the 16S rRNA gene was amplified using primers for the 515F-806R with barcodes for each sample and Illumina sequencing adapters (*Caporaso et al., 2011*). Three independent PCR amplifications were done for each DNA sample with the following program: 98 °C for 30 s followed by 35 cycles of 95 °C for 30 s, 52 °C for 40 s, and 72 °C for 90 s, and a final elongation step of 12 min at 72 °C, then kept at 4 °C. PCR products were pooled for each sample and purified with the AmpliClean™ Magnetic Bead PCR (NimaGen, NDL) system. PCR amplicon libraries were quantified using a QUBIT fluorometer (Promega, Madison, WI, USA). A total of 20 ng/$\mu$l per sample was used for sequencing on a 2 × 300 Illumina MiSeq platform (Yale Center for Genome Analysis,

CT, USA). Sequences generated in this study are deposited under BioProject number PRJNA683724.

Sequences were processed using the Quantitative Insights into Microbial Ecology 2 (QIIME 2, v.2018.6) pipeline (*Bolyen et al., 2018*). Data were filtered and denoised using the dada2 pipeline (*Callahan et al., 2016*). Sequences were assigned into ASVs (Amplicon Sequence Variants), discarding merged reads shorter than 200 bp and an average quality score of >25. Sequences were aligned using MAFFT (*Katoh & Standley, 2013*), and highly variable regions were masked from the alignment. A phylogenetic tree was constructed using FastTree (*Price, Dehal & Arkin, 2009*) to describe the microbial composition for each lake and sampled depth. ASVs were assigned a rank-based taxonomy using vsearch v1.9.5 (*Rognes et al., 2016*) and SILVA database release 132 at 99% similarity for the 515-806 region (*Quast et al., 2012*). A Neighbor-Joining phylogenetic tree was constructed to classify undefined taxa at the genus level (Fig. S1). Sequences were analyzed using multivariate correlational and ordination methods in the R environment (version 3.6.2). For this, we used Phyloseq R (*McMurdie & Holmes, 2013*), ggplot2 (v 2.1.0) (*Ginestet, 2011*), ampvis2 and vegan packages (v 2.3–5) (*Oksanen et al., 2013*). ASVs representing less than 1000 of the sequences across the dataset, chloroplast, and mitochondrial sequences were not considered in this analysis.

Alpha diversity metrics, including Observed ASVs, Chao1 index, Shannon, and Simpson, were estimated for each lake (Table S4). Based on Bray-Curtis dissimilarity, Canonical Analysis of Principal Coordinates (CAP) (*Anderson & Willis, 2003*) was used to evaluate the correlations between microbial composition and diversity with environmental variables. Statistical analysis of the microbial composition and diversity of the different lakes included a PERMANOVA (Adonis function in R vegan package) (*Zapala & Schork, 2006*) to evaluate the effect of depth, anoxic/anoxic conditions, and other environmental variables. A $p$-value <0.05 was considered significant allowing for 1000 permutations. All the indices for alpha diversity were compared using tests of Mann–Whitney–Wilcoxon; a value of $p < 0.05$ was considered a positive result.

# RESULTS

## Physicochemical characterization

Eutrophic lakes were by far more turbid, with higher mineralization and Chl-a concentration than oligotrophic lakes, which were transparent, less mineralized, and with lower Chl-a concentration (ultraoligotrophic/oligotrophic) (Table 1). The two groups of lakes had statistically differences ($p < 0.05$) in $Z_{SD}$, $Z_{EU}$, $K_{25}$, and Chl-a concentrations and were similar in temperature, pH, and DO. The physical and chemical characteristics per lake are provided (Table S3 raw data).

There was a clear difference in water transparency between the eutrophic lakes -turbid- and the oligotrophic lakes -crystal clear water. The origin of the turbidity in the eutrophic lakes is biological (phytoplankton) and not terrigenous (clays), as confirmed by the larger concentration of Chl-a in the eutrophic lakes, which was approximately 40 times higher than in the oligotrophic lakes.

**Table 1  Physical and chemical characteristics of the two main lake-types of the LMNP.**

| Plateau/ Eutrophic | $Z_{SD}$ (m) | $Z_{EU}$ (m) | Temp (°C) | DO (mg L$^{-1}$) | DO (% sat) | pH (U) | $K_{25}$ ($\mu$S cm$^{-1}$) | Chl-a ($\mu$g L$^{-1}$) |
|---|---|---|---|---|---|---|---|---|
| X | 0.7 | 2.2 | 18.4 | 1.2 | 17.7 | 7.2 | 512 | 31.9 |
| s.d. | 0.5 | 0.4 | 0.7 | 2.8 | 41.8 | 0.4 | 85 | 29.3 |
| Min | 0.4 | 1.8 | 17.7 | 0 | 0 | 6.8 | 429 | 1.6 |
| Max | 1.3 | 2.7 | 21.7 | 12 | 177.4 | 8.5 | 691 | 86.7 |
| **Mountain/Oligotrophic** | | | | | | | | |
| X | 8.7 | 30.4 | 19.2 | 4.6 | 66.6 | 7.6 | 333 | 0.8 |
| s.d. | 4.1 | 15.4 | 1.1 | 3.4 | 49.7 | 0.6 | 202 | 0.9 |
| Min | 3.2 | 20.5 | 17.5 | 0 | 0 | 6.3 | 187 | 0.1 |
| Max | 16.2 | 56 | 24.2 | 8.1 | 121.3 | 8.4 | 1,483 | 6.1 |

Notes.

$Z_{SD}$, Secchi depth; $Z_{EU}$, euphotic layer; Temp, temperature; DO, dissolved oxygen; %sat, saturation percentage; U, pH units; $K_{25}$, electrical conductivity at 25 °C; Chl-a, chlorophyll-a concentration; X, average; s.d., standard deviation; Min, minimum; Max, maximum.

Temperature profiles of the LMNP showed the presence of scattered depths along the water column with thermal gradients $\geq 0.3$ °C m$^{-1}$ except for Pojoj and Cinco Lagos, which were quite homothermic (thermal gradients $<0.3$ °C m$^{-1}$). Thermal gradients $\geq 0.3$ °C m$^{-1}$ were found in two different scenarios, including (1) three eutrophic and some oligotrophic lakes (*e.g.*, Cinco Lagos, Kichail) that started at the surface, where there was no epilimnion, which might be inferred as a transient thermal feature (intense solar radiation, no wind stress); (2) in oligotrophic lakes, even though there are some depths with thermal gradients $\geq 0.3$ °C m$^{-1}$, which do not constitute a well-defined layer (metalimnion).

Even though temperature profiles suggested all lakes were circulating, DO profiles showed otherwise. We found deep water anoxia in nine out of 12 lakes revealing incomplete water column mixing. Eutrophic lakes had the highest Chl-a concentrations on the surface, diminishing with depth. In oligotrophic lakes, Chl-a was uniform along the water column in those lakes already circulating or concentrated at mid- to deeper water (*i.e.*, DCM, deep chlorophyll maximum) in those lakes yet to be in full circulation.

A Dendrogram constructed with the physical and chemical characteristics of the sampled lakes (Fig. 3A) showed three groups (with ten of dissimilarity distance). The lakes of the first group (A) included only oligotrophic lakes Montebello, Tziscao, Ensueño, Agua Tinta, and Kichail. The second group (B) included eutrophic Bosque Azul, La Encantada, and San Lorenzo, with oligotrophic Esmeralda and Dos Lagos. The third group (C) included oligotrophic Cinco Lagos and Pojoj.

In the PCA (Fig. 3B), the first two principal components (PC) explained nearly 98% of the total variance. For PC1, $K_{25}$ was the most heavily loaded variable (2.26). The lakes with the lowest $K_{25}$ are in the left part of the PC1. The $Z_{EU}$ (2.06) and Chl-a ($-1.16$) had the most significant influence on PC2. The highest values of $Z_{EU}$ are in the positive part of the PC2, while the highest values of Chl-a are in the negative part of PC2. The same three groups revealed by the cluster analysis were identified in the PCA.

Group A (Tziscao, Ensueño, Agua Tinta, Kichail, Montebello) includes lakes with lower values of $K_{25}$ and Chl-a concentration and higher $Z_{EU}$; Group B (Bosque Azul, La

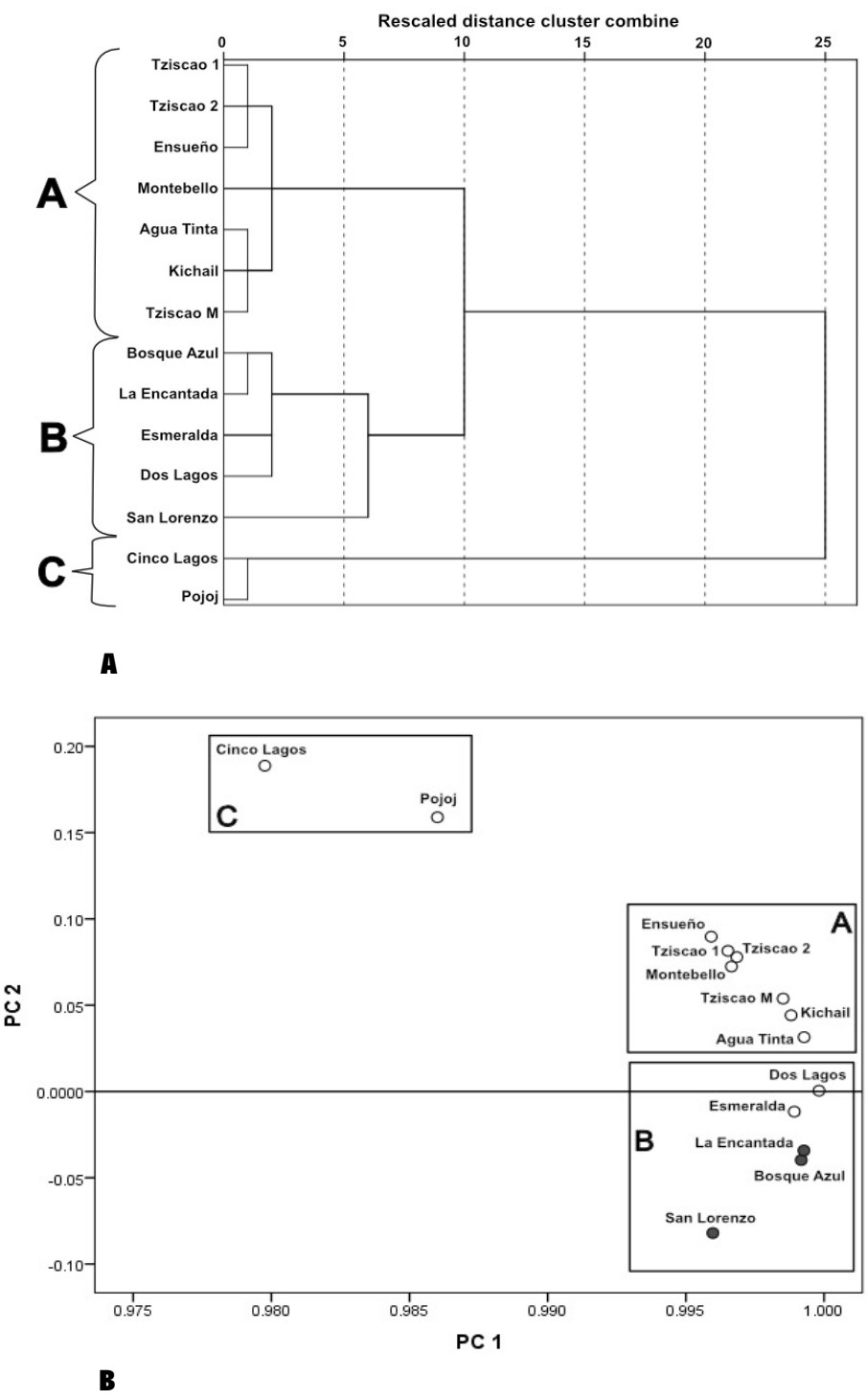

**Figure 3** **Cluster analysis dendrogram (A) and PCA (B) based on the physical and chemical characteristics of the Montebello lakes.** (● plateau/eutrophic, ○ mountain/oligotrophic).

Encantada, Esmeralda, Dos Lagos, San Lorenzo) includes lakes with highest values of $K_{25}$ and Chl-a concentration, and the lowest $Z_{EU}$. Group C (Cinco Lagos, Pojoj) contained lakes with the lowest $K_{25}$ and Chl-a concentration values and the highest $Z_{EU}$.

## Water column microbial composition and diversity

A total of 15, 213, 405 paired-end V4 16S rRNA sequences were recovered in this study. After quality filtering, a total of 10,744,186 sequence reads remained in the dataset, which consisted of 27,319 ASVs identified based on single nucleotide variations in the sequence reads. All lakes included in this study were equally sampled, and all collector curves reached an asymptote for sequence reads. Alpha diversity metrics based on ASVs were similar between lake conditions (Tables S4 and S5). The sequences obtained in this study mainly represented different phyla within the Bacteria domain, where Proteobacteria and Cyanobacteria were dominant (>50% and up to 98%). There was no correlation between the lakes' trophic, aerobic, or anaerobic conditions and microbial diversity, although the abundance of specific taxa did vary. On average, microbial diversity increased with depth. Eutrophic lakes (San Lorenzo, La Encantada and Bosque Azul) had anaerobic conditions along most of the water column (particularly, La Encantada). These lakes were dominated by Cyanobacteria, Proteobacteria, and Bacteroidetes, where at the genus level, the filamentous cyanobacteria *Planktothrix* (Order Oscillatoriales, Family Phormidiaceae) (Figs. 4 and 5) comprised up to 97% of all cyanobacterial sequences in shallow depths, which were closely affiliated to *P. agardhii* (Fig. S1). Other phyla included Actinobacteria (up to 40%), Verrucomicrobia (up to 12%), Bacteroidetes (up to 25%), Firmicutes (>5%), and Fusobacteria (>2%) (Fig. 4). The eutrophic lakes showed a pattern where Cyanobacteria and Bacteroidetes (including Lentimicrobiacea, an anaerobic bacterium) had a higher percentage in the composition compared to oligotrophic lakes where Proteobacteria was predominant. At the Family level, the microbial groups' separation is associated with the trophic state categories (Fig. 5). Archaea represented 1.5% of the entire community: Nanoarchaeota was the most abundant phylum in eutrophic lakes, whereas Thaumarchaeota and Euryarchaeota were more abundant in oligotrophic lakes (Figs. 4 and 5). In oligotrophic lakes, *Cyanobium*, a unicellular cyanobacterium, was the dominant microbe between 10–15 m, and Moraxellaceae, a mesophilic Gammaproteobacterium dominated between 10 to 30 m. Mainly, Ensueño, an oligotrophic lake, had Chlamydiae in most depths; Cinco Lagos Pojoj and Dos Lagos, also oligotrophic, had Omnitrophicaeota; Ensueño, Tziscao, and Pojoj were the only lakes with Acidobacteria, while Tziscao was the only lake with Tenericutes, and with Pojoj, harbored Rokubacteria.

According to the temperature and DO profiles, the water column of the lakes was mostly or entirely mixed, which was also evident in the microbial composition, where no significant differences were observed along the water column (Table S4). There are significant differences (Adonis test, $p > 0.05$) between eutrophic and oligotrophic lakes ($p > 0.05$) (Fig. 6, Tables S4 and S5), and Chl-a concentration was the main explanatory variable (Fig. 6).

Group A of the environmental cluster analysis (Fig. 3A) comprised the oligotrophic lakes with the lowest Chl-a concentrations. These lakes had a microbial composition dominated
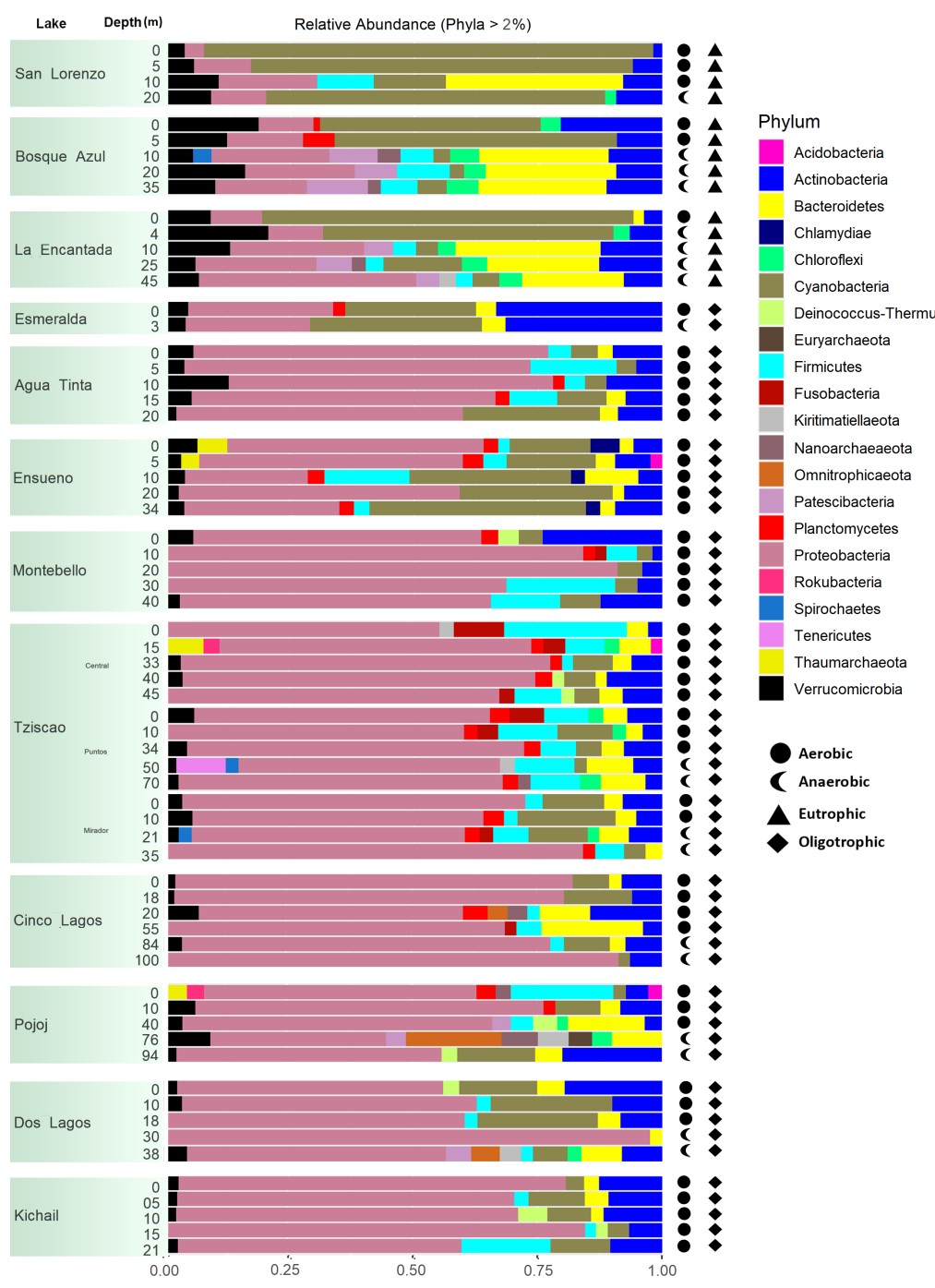

**Figure 4 Bar plots showing microbial composition and diversity of LMNP water column assemblages at the phylum (A) and family (B) levels.** Different depths per lake and aerobic/anaerobic and eutrophic/oligotrophic categories are shown.

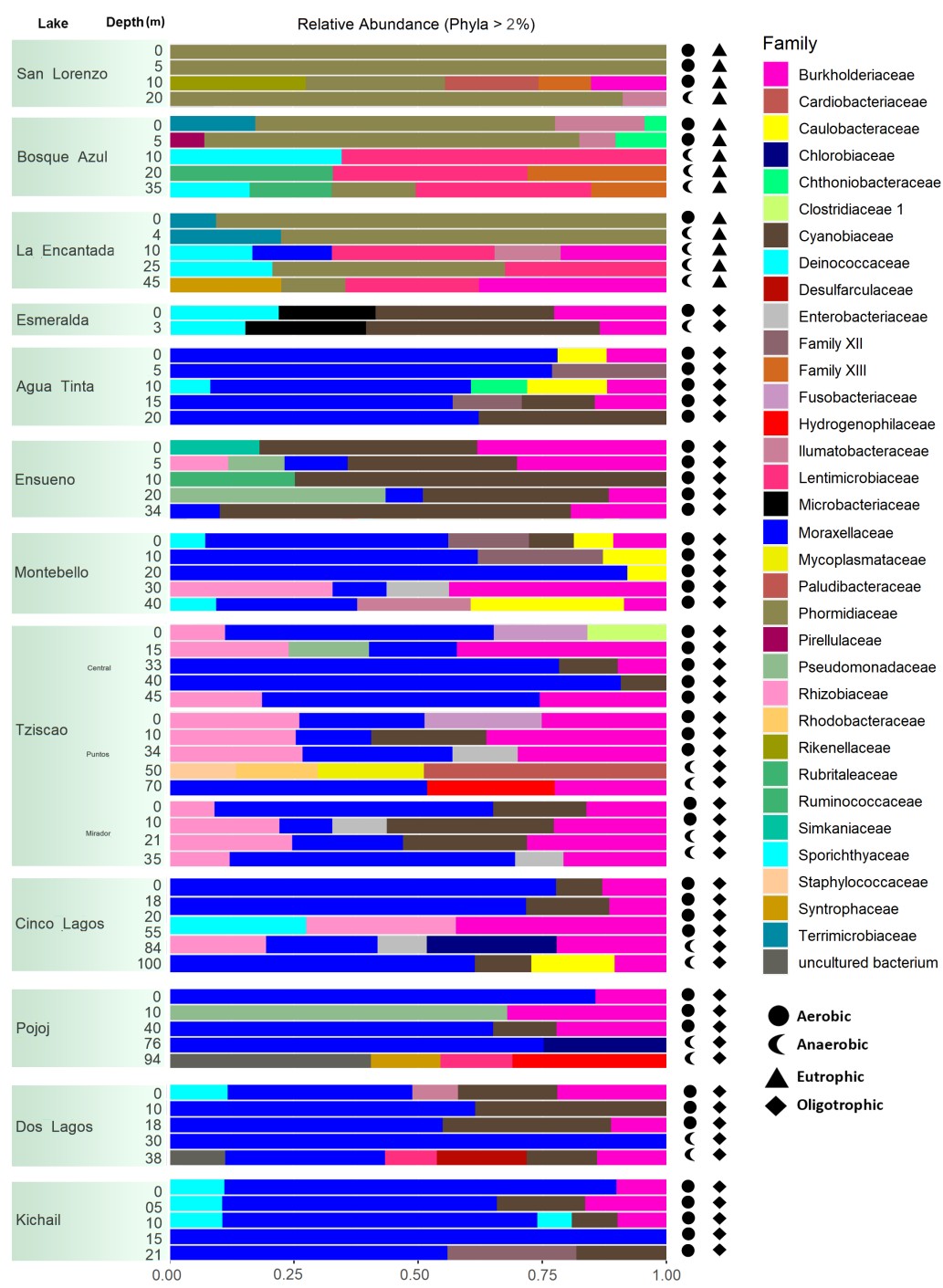

**Figure 5** Bar plots showing microbial composition and diversity of LMNP water column assemblages at the phylum (A) and family (B) levels. Different depths per lake and aerobic/anaerobic and eutrophic/oligotrophic categories are shown.

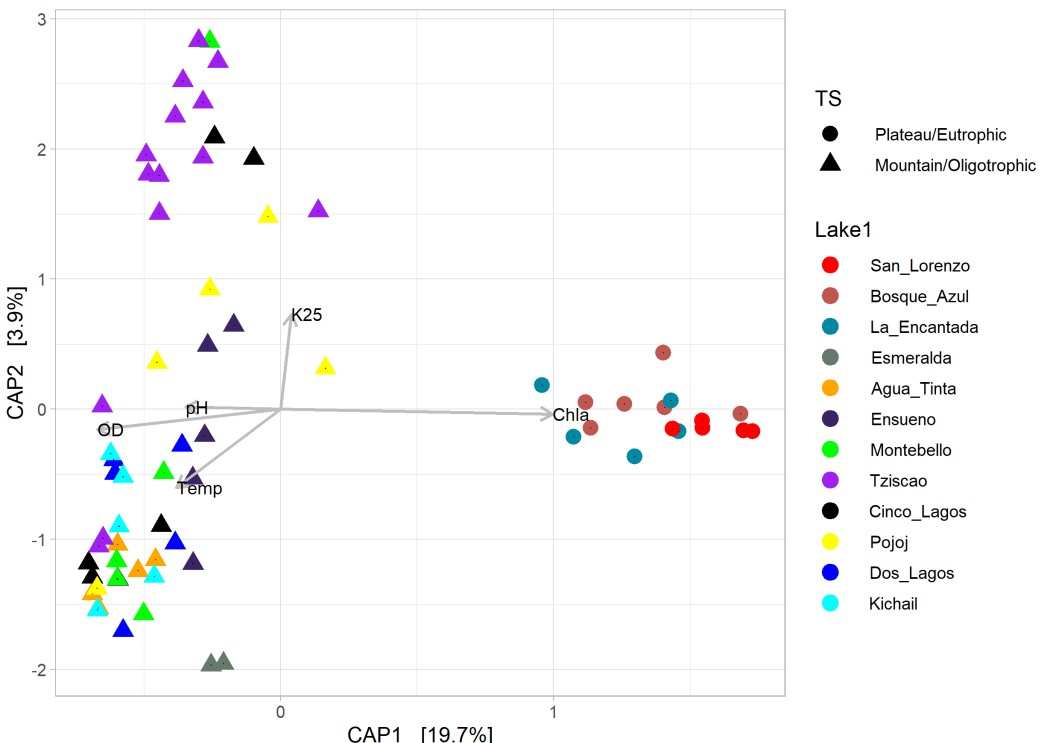

**Figure 6** CAP ordination showing the grouping of the lakes included in this study considering the environmental and genetic datasets in two main categories: mountain/oligotrophic.

by Acinetobacter (24–51%) and the unicellular cyanobacterium *Cyanobium* (2.4–14%), with Ensueño showing a clear dominance of 32% for this picocyanobacteria and 4.2% of Acinetobacter. Montebello was the only lake within the oligotrophic group that had *Planktothrix* (9.7%), which also raises concerns regarding its trophic evolution (Fig. 6). Cinco Lagos and Pojoj grouped with the oligotrophic lakes regarding their microbial composition and diversity (Figs. 6 and 7) yet are a separate group regarding environmental variables (Fig. 3A and 3B) since both lakes are ultra-oligotrophic. There, Acinetobacter was the dominant bacteria (31–40%), *Cyanobium* was the dominant cyanobacteria, and *Achromobacter, Rhizobium*, and *Brevundimonas* were also present.

This analysis suggests a specific type of microbe, in this case, the filamentous, toxin-producing cyanobacteria *Planktothrix*, dominates in eutrophic lakes. Differently, the oligotrophic lakes *Acinetobacter* sp. and *Cyanobium* dominate. The deep sequencing approach applied in this research generated results that suggest that the lakes of the LMNP have similar microbial diversity (Table S4) but different compositions (Fig. 7, Table S5). Two main groups of cyanobacteria found in the lakes of the LMNP are bioindicators of trophic states: *Planktothrix* of eutrophic and *Cyanobium* of oligotrophic environments (Fig. S1). In the oligotrophic lakes, *Cyanobium* represented up to 80% of all the cyanobacteria, which also included the picocyanobacteria *Synechococcus* (5%) and *Microcystis* (1%), and

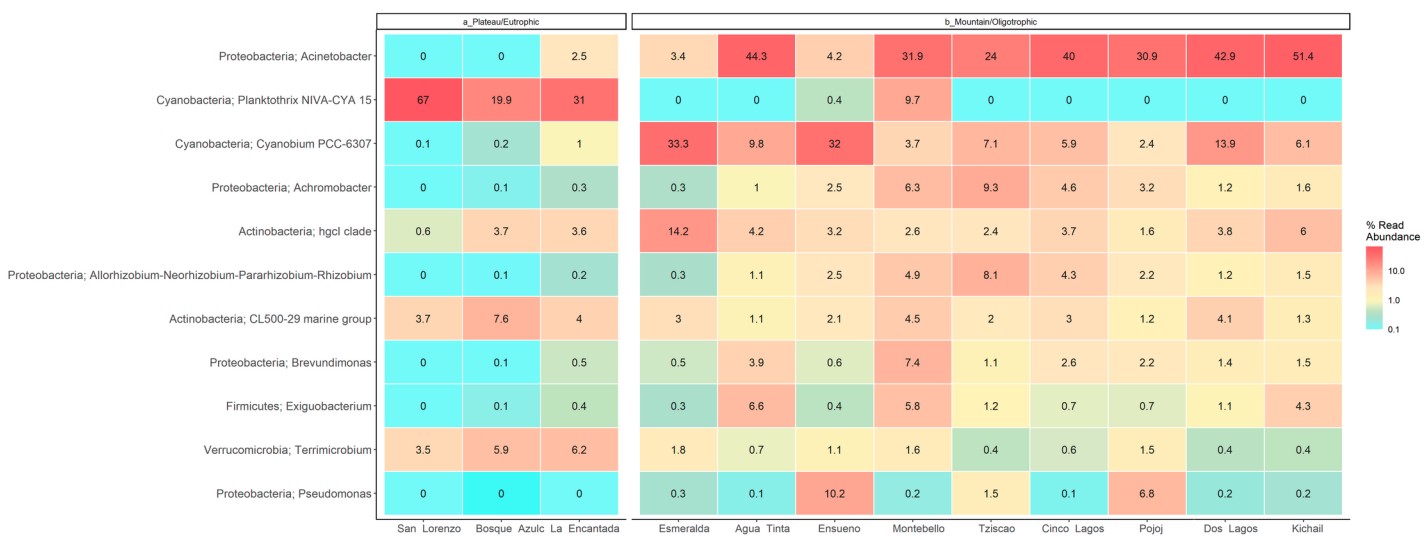

**Figure 7** LMNP heat map showing microbial diversity at genus level for plateau/eutrophic and mountain/oligotrophic lakes.

the filamentous *Nodosilinea* (1%), among others. The most abundant genus in all eutrophic lakes was the filamentous cyanobacteria *Planktothrix* (Fig. 7).

## DISCUSSION

### Physicochemical characterization of the lakes of the LMNP

The eutrophication of lakes has become a global water pollution problem, and Chl-a, total nitrogen (TN), total phosphorus (TP), chemical oxygen demand (COD), and $Z_{SD}$ are the leading indicators to evaluate their level of eutrophication (*Du et al., 2019*; *Wurtsbaugh, Paerl & Dodds, 2019*). Sampling campaigns for this study took place in late winter when the lakes should have experienced full mixing. Further, homeothermic profiles indicate no density barriers were preventing the water column from mixing; nonetheless, some DO profiles showed deep water anoxia. If deep water anoxia from the stratification period remains, then the water column mixing has not taken place yet since mixing requires the kinetic energy of wind to circulate. It must be noted that: (a) warm-monomictic lakes circulate in the hemispheric winter (*Lewis Jr, 1996*), and (b) tropical thermoclines could be fully functional with temperature gradients $\geq 0.5\,°C\,m-1$ or even $\geq 0.3\,°C\,m-1$ (*Pérez & Restrepo, 2008*).

Chlorophyll is present in all photosynthetic phytoplanktonic organisms, and, therefore, its concentration has traditionally been used to estimate phytoplankton biomass in lakes and oceans (*Brown, 1977*). Generally, Chl-a concentration has a positive correlation in water bodies associated with eutrophication processes and blooms (*Pérez-Ruzafa et al., 2019*; *Quinlan et al., 2021*). The vertical distribution of Chl-a in this study is also different in both lake types, as had been already reported (*Rivera-Herrera et al., 2019*; *Vera-Franco et al., 2015*). In the eutrophic lakes, the highest Chl-a concentrations are in the upper layer, close to the surface, diminishing downwards. Differently, in the oligotrophic lakes,

the Chl-a distribution is either more or less uniform along the water column in those lakes already circulating or concentrated at mid-water (*i.e.*, DCM) or from mid- to deeper waters in those lakes yet to be in full circulation. DCM are characteristic features of oligotrophic waters (*Cullen, 1982*; *Fee, 1976*). Although research in other study sites has shown that eutrophication increases towards the summer when stratification reaches its maximum, *Vargas-Sánchez, Alcocer & Oseguera (2022)* found higher Chl-a concentrations in the lakes of the LMNP during the cold/dry season (winter) than in the warm/rainy season (summer). Tropical warm monomictic lakes show characteristic winter phytoplankton blooms associated with the full water column mixing (*Lewis Jr, 1996*; *Sarmento, 2012*).

Besides Chl-a concentration, indicators of water transparency ($Z_{SD}$, $Z_{EU}$) evidenced the sample differences between the turbid (green) eutrophic and the transparent (blue) oligotrophic lakes (Fig. 2). Since Chl-a and water turbidity were the main environmental parameters associated with the trophic state of the lakes, these variables could be monitored through satellite imaging in the future (AQUA-MODIS), using gradient boosted decision trees (GBDT) and regression models that allow the quantitative evaluation of such parameters remotely (*Allan et al., 2011*; *Germán et al., 2020*; *Lazcano et al., 2018*), allowing long-term monitoring of the trophic status of the lakes.

Eutrophication in the lakes of the LMNP was first noted in 2003, associated with the growing anthropogenic activities along the Río Grande de Comitán river, regional urban development, untreated sewage delivery, changes in land use, and transformation of forested into agricultural areas in zones next to the LMNP (agricultural run-off), which have been the main problem since 1940 (*Alcocer, 2017*; *Caballero et al., 2020*). Nonetheless, long-term efforts to understand these lakes' seasonal changes and magnitude of eutrophication are needed.

## Microbial composition in the water column of the LMNP

Over 27,000 ASVs were recovered from more than 10 million sequence reads for the water column characterization of different lakes in the LMNP. Most of these ASVs were present in all lakes, hence the similarity in bacterial diversity along the water column and trophic states. Results clearly show that certain bacteria thrive and are conducive to eutrophic conditions, increasing their abundance significantly (*e.g.*, *Planktothrix*). This deep sequencing approach also allowed for the recognition of specific conditions in the lakes. For instance, the correlation analysis of environmental variables (group B, Fig. 3) included oligotrophic Dos Lagos and Esmeralda with three eutrophic lakes. However, their microbial diversity and composition (Tables S4 and S5) are similar to oligotrophic lakes (Figs. 4–6). Located in the transition zone between the plateau and mountain lakes, the Chl-a concentration in both lakes is far below that of the eutrophic lakes ($31.9 \pm 29.3$ µg $L^{-1}$) but similar (Esmeralda: $0.5 \pm 0.1$ µg $L^{-1}$) or slightly higher (Dos Lagos: $1.9 \pm 2.4$ µg $L^{-1}$) than the oligotrophic lakes ($0.8 \pm 0.9$ µg $L^{-1}$). In addition, Dos Lagos shares with eutrophic lakes a high $K_{25}$, while Esmeralda shares with eutrophic lakes a low $Z_{EU}$. However, the high salinity in Dos Lagos derives from saline, gypsum-rich groundwater (*Alcocer, 2017*) and not from water pollution as in eutrophic lakes. The low $Z_{EU}$ in Lake

Esmeralda and a low Chl-a concentration indicate turbidity is not biogenic like in eutrophic lakes but terrigenous.

Several studies have reported that *Cyanobium* and *Synechococcus* are the dominant picocyanobacteria in oligotrophic or mesotrophic freshwater lakes (*Callieri, 2008*; *Komárek, 1996*). *Cyanobium,* like other picocyanobacteria, plays an important role as primary producers in oligotrophic freshwater and marine ecosystems (*Guillou et al., 2001*). *Planktothrix* is one of the dominant cyanobacteria in eutrophic lakes. The presence of toxin-producing cyanobacteria, *e.g.*, *P. agardhii* and *P. rubescens* (*Davis et al., 2015*; *Yéprémian et al., 2007*), is known to be associated with eutrophic freshwater bodies and deterioration of water quality (*Westrick et al., 2010*). Hence, *Planktothrix* can be used as an indicator species of eutrophic conditions since extensive information is available on its distribution and ecology (*e.g.*, temperature, light, nutrient requirements) (*Bonilla et al., 2012*). Recently, a study reported the prevalence of *Planktothrix* in eutrophic lakes of the LMNP (*Fernández, Alcocer & Oseguera, 2021*) and signaled their capacity to produce cyanotoxins. *Planktothrix* can competitively exclude other photosynthetic microorganisms while blooming (*Kormas et al., 2011*), which is most likely why it dominates the eutrophic lakes in the LMNP (*Fernández, Alcocer & Oseguera, 2021*). Overall, the most abundant Proteobacteria in the lakes of the LMNP was *Acinetobacter* representing 31% of all sequences. *Acinetobacter* is a highly competitive microorganism that controls cyanobacterial and microalgal blooms (*Su et al., 2016*; *Wang, Li & Kang, 2007*). However, their greater abundance in oligotrophic lakes and a significant decrease in eutrophic lakes of the LMNP suggest *Planktothrix* outcompetes it.

The attributes of the microbial components of the aquatic communities, including their composition and diversity, respond to environmental changes according to spatial and temporal characteristics (*Gough & Stahl, 2011*; *Vander Gucht et al., 2005*). Hence, studies need to include frequent sampling periods along the whole circulation-stratification seasons to recognize the environmental and microbial dynamics of the studied systems. More data and continuous studies that include deep sequencing approaches coupled to the environmental characterization of the systems are needed to understand the network of biological interactions that define the composition of aquatic ecosystems.

In recent years the characterization of the microbial component in the environment has been made possible using massive sequencing of genetic regions, including the *16S rRNA* gene (*Ruuskanen et al., 2018*; *Vander Gucht et al., 2005*; *Zhang et al., 2018*). In this study, molecular tools helped to gain insight and evaluate the microbial composition and diversity in the water column assemblages analyzed in lakes with different trophic conditions (*Garlapati et al., 2019*). A perturbation in ecological systems such as eutrophication affects the structure of communities and changes their composition, favoring the dominance of some species, as reported in this study. Montebello, one of the largest lakes in the LMNP, needs to be closely monitored since the analysis discussed here suggests its composition is starting to shift towards eutrophic conditions, even though the environmental parameters classify it as oligotrophic. Further, the microbial composition is one of the first components to respond to environmental disturbances; hence, monitoring the microbial communities in the water column of LMNP may be a surveillance strategy that allows us to learn more

about the eutrophication process and its effects. We consider that new approaches that couple ecosystem health monitoring and sustainable policies are necessary to predict future scenarios in freshwater ecosystems, including LMNP.

Concluding, the shift from oligo to eutrophic conditions resulted in (1) higher turbidity, mineralization, and Chl-a concentrations; (2) the replacement of *Cyanobium* and *Synechococcus* as dominant cyanobacteria in the water column by the cyanotoxin-producing *Planktothrix*; (3) the microbial diversity remained similar between eutrophic and oligotrophic lakes, but there were significant changes in composition.

## ACKNOWLEDGEMENTS

Technical support is acknowledged to Osiris Gaona (IE, UNAM) and Joanna Ortiz (F. Ciencias, UNAM, PCTY).

### Funding
This work was supported by the Dirección General de Asuntos del Personal Académico (DGAPA-PAPIIT) through projects IN219215 "Factores que determinan el estado trófico de los lagos de Montebello, Chiapas", IV200319 "Área Experimental de Lagos Tropicales", Instituto de Ecología, IV200122 (Javier Alcocer), UNAM and PAPIIT IN207220 (LIF). Alfredo Yanez-Montalvo, Bernardo Aguila, and Miriam Guerrero-Jacinto received graduate studies scholarships from CONACYT, México. Elizabeth Selene Gómez-Acata received a postdoctoral scholarship from UNAM-DGAPA. The funders had no role in study design, data collection and analysis, decision to publish, or preparation of the manuscript.

### Grant Disclosures
The following grant information was disclosed by the authors:
Dirección General de Asuntos del Personal Académico (DGAPA-PAPIIT): IN219215.
Factores que determinan el estado trófico de los lagos de Montebello, Chiapas: IV200319.
'Área Experimental de Lagos Tropicales", Instituto de Ecología: IV200122.
UNAM and PAPIIT: IN207220.
CONACYT, México.
UNAM-DGAPA.

### Competing Interests
The authors declare there are no competing interests.

### Author Contributions
- Alfredo Yanez-Montalvo conceived and designed the experiments, performed the experiments, analyzed the data, prepared figures and/or tables, and approved the final draft.
- Bernardo Aguila conceived and designed the experiments, performed the experiments, analyzed the data, prepared figures and/or tables, and approved the final draft.

- Elizabeth Selene Gómez-Acata performed the experiments, analyzed the data, prepared figures and/or tables, and approved the final draft.
- Miriam Guerrero-Jacinto analyzed the data, prepared figures and/or tables, and approved the final draft.
- Luis A. Oseguera performed the experiments, analyzed the data, prepared figures and/or tables, and approved the final draft.
- Luisa I. Falcón conceived and designed the experiments, performed the experiments, analyzed the data, prepared figures and/or tables, authored or reviewed drafts of the article, and approved the final draft.
- Javier Alcocer conceived and designed the experiments, analyzed the data, prepared figures and/or tables, authored or reviewed drafts of the article, and approved the final draft.

## Data Availability

The data is available at NCBI: PRJNA683724.

## Supplemental Information

Supplemental information for this article can be found online at http://dx.doi.org/10.7717/peerj.13999#supplemental-information.

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
