# Peer review of "Shifts in water column microbial composition associated to lakes with different trophic conditions: “Lagunas de Montebello” National Park, Chiapas, México"

_PeerJ, doi:10.7717/peerj.13999_

## Round 0.1 · original submission · Major Revisions

This article is a solid submission to PeerJ but the writing is hindering its acceptance. Even with revision please be aware that the manuscript may not eventually be accepted: it is critical that it be thoroughly revised for readability and interpretation. Read the excellent reviews provided and carefully consider their recommendations. Both reviewers understood the quality and value of the research, but both were also quite critical of the narrative structure and overall writing. I sternly encourage you to do extensive revision of the results, discussion and overall flow of the manuscript in a revision, specifically addressing the points raised by the reviewers in the process.

Reviewer 1 ·

Basic reporting

In this manuscript, the authors compared physicochemical parameters and microbial community composition in twelve lakes out of which nine were oligotrophic and three were eutrophic. The authors found that chlorophyll a content was the main environmental parameter correlated with lake trophic status. They also reported clear differences in microbial community composition between eutrophic and oligotrophic lakes. In general, this study provides interesting microbial data of freshwater lakes. The overall setting of this work also allows comparative studies within a limited geographical range. However, I have a couple of critical points regarding the experimental design, display and interpretation of results that I suggest the authors should pay attention to.
In general, it remains unclear what is the main message/outcome of the study and which gap of knowledge is being addressed. Here, I would recommend that the authors address these points more strongly throughout the manuscript. In large parts, the discussion also remains on a rather general level (e. g., l. 405-411; third conclusion l. 422-423). Here, a more in depth discussion of the data obtained in this study and how they do not only confirm previous findings but also add new aspects would be desirable. In this context, the data presented in this work have a much higher potential than is being exploited in the current version of this manuscript. For example, Fig. 4a is a really nice figure with a lot of information. However, only very little of this is eventually taken up in the results and discussion section. I would also recommend that the authors make more use of the vertically resolved physicochemical data that they generated, i. e. individual (supplemental) figures in addition to the presentation of mean values.

Experimental design

The major aspect I would like to bring up here is the definition of oligotrophic versus eutrophic lakes. The introduction provides a sound overview of the environmental problems associated with lake eutrophication and potential drivers of this eutrophication. However, I would recommend the authors differentiate a bit more between “eutrophic” as a trophic status and the process of eutrophication. Under natural conditions, most oligotrophic lakes will eventually turn into eutrophic lakes by natural succession, except for specific climatic regimes or high altitudes. Consequently, the eutrophic state is not bad as such and often associated with a higher diversity and productivity of macrophytes, planktonic organisms, and fish. Of course, preservation of the oligotrophic status of lakes is desirable for multiple reasons. I understand the authors focus more on eutrophication processes enhanced by human activities, where lake trophic status may also go beyond the eutrophic state, that is become hypertrophic or polytrophic. I would recommend to phrase this more precisely where needed, and also to restructure the introduction slightly so that the reader is guided more directly towards this aspect.

Validity of the findings

It is also not fully clear on which facts the original classification of these lakes as oligotrophic or eutrophic is based. In the conclusions, the authors identify turbidity, mineralization and chlorophyll-a concentrations as main indicators of eutrophication. But it is not clear whether the original classification of lakes as stated in the method section (l. 116-126) was already based on exactly these parameters. It also appears that circulation dynamics of the different lakes are indirectly linked to lake trophic status because the oligotrophic lakes happen to be located in places less exposed to wind mixing compared to the eutrophic lakes. Here, the authors need to be more careful when interpreting their results, because this indirect link might also affect potential other relationships between lake trophic status and chemical parameters and their vertical distribution.

Additional comments

l. 122-126: Why did the authors contrast 3 eutrophic lakes with 6 oligotrophic lakes?
l. 147 ff: I suggest the authors have one combined section covering all the statistical analysis. In the current version of the manuscript, the statistical analysis done on the sequencing data is integrated into the section explaining bioinformatics analysis.
l. 163-172: Please add information here how samples were prepared for Illumina sequencing, that is at what step were adapters and barcodes introduced? Already in the first PCR step?
l. 188: To what threshold of relative abundance do these 1000 sequences correspond?
Table 1: What are these data based on? Is this mean values obtained from different water depths? Please explain.
l. 210ff: Usually, text in the results section is written in the past tense. Please rephrase.
l. 215-227: It is unclear to what extent comparisons between the different lakes are affected by the fact that lakes were in different stages of circulation by the time point of sampling. This is not necessarily driven by trophic status but rather location in the landscape, wind exposure etc. Here, the authors should comment on if comparability of results across lakes might be affected by the different stages of water column mixing.
l. 295-299: Given the fact that samples were only taken at one time point from the different lakes, the authors need to be more careful with generalizations regarding microbial community composition. This may also vary across seasons in a lake-dependent manner, and the mentioned microbial taxa might have a different indicatory value at other times of the year.
L. 318-320: Is not fully clear why authors did only one sampling in winter to assess parameters representative of the status of eutrophication. TN and TP might act as clear indicators in winter when primary production is low. But would the Chl-a content not be a better indicator during the summer period when primary production is high?
L. 322-325: This section is not clear. Did all lakes represent a state of complete mixing, or did they not? Please rephrase.
l. 337-342: In this line of argumentation, the effects of lake trophic status and circulation status are getting mixed. It is unclear if the authors here also aim to link chlorophyll distribution to lake trophic status?
l. 345-349: To what extent would satellite based analysis also allow the resolution of depth-dependent distribution of Chl-a and turbidity? Please explain.
l. 354-360 and l. 399-404: I agree with the authors that long-term efforts are needed to understand seasonal changes. But the fact that only winter samples were available also limits the conclusions that can be drawn from the results presented in this manuscript.
l. 364-365: This sentence is unclear, please rephrase.
l. 370-377: The main message of this paragraph is unclear. Is it that microbial parameters are a better indicator of lake trophic status than the chemical parameters? Or is it a critical discussion of the suitable of the chemical indicators used in this study? The latter might be better placed along with the discussion of the physicochemical parameters in the sections above.
l. 393-396: Why would Acinetobacter be a better competitor than Planktothrix in these lakes? What are the assumed metabolic and ecological functions of Acinetobacter? Please explain.
Figure 1: The legend is not fully clear. What is the difference between “forest”, “rainforest” and “vegetation”?
Figure 3a/3b: It seems there is some confusion in the numbering of figures here (fig. 3b vs fig. 4).
Fig. 4a: I assume depth is in m? Please specify.
Figure 6 could also be shown as Supplemental Material

Reviewer 2 ·

Basic reporting

Overall, the authors propose an important question pertaining to water quality in a protected natural area that is threatened by agricultural runoff. This research paper was very interesting to read. The authors’ research questions aligned with their objective, it is clear and concise. Their intent to identify water quality conditions and microbial community is derived from Mexico’s decree to protect Lagunas de Montebello National Park area. This area is surrounded by agricultural landuse, which affects trophic status of these lakes and potentially lead to changes in water quality and microbial community. While readers who are interested in water quality will understand the "why" of this research, readers who are not familiar with water quality (or are new to water quality) may not grasp the "why" of this research.

As a reader, it is easy to identify where the authors are having trouble. There are several abrupt changes in the transitions and readers may need to re-read the sentence again to grasp what the authors are trying to conveyThe authors are exploring water quality conditions and microbial community during the winter season in a set of lakes that comprise of both oligotrophic and eutrophic conditions. Their site selection is appropriate for their research question, as it gives insight into the lake conditions and microbial community in both eutrophic and oligotrophic conditions during the winter season. The methods used are appropriate and there are a few minor areas that need further clarification in their methods..

Experimental design

The authors are examining how water quality parameters and microbial community changes from oligotrophic to eutrophic conditions one of Mexico’s protected natural areas, which receives surface runoff from agricultural landuse. Their site selection is appropriate for their research question, as it gives insight into the lake conditions and microbial community in both eutrophic and oligotrophic conditions during the winter season. The methods used are appropriate and there are a few minor areas that need further clarification in their methods. Because sampling was conducted in the winter, it is hard to know if conditions remain the same during the summer season. It would be interesting to see whether some of the lakes transition from oligotrophic status to eutrophic status during the summer season.

Validity of the findings

The results from the 16S rRNA sequence dataset are interesting, as they reveal differences in microbial composition in oligotrophic vs. eutrophic conditions. The presence of a toxin producing cyanobacteria in oligotrophic lake does raise concern that these lakes could potentially become eutrophic in the future. However, it can be hard for readers to follow the results because Table 1 only shows the average, min, max, and standard deviation of eutrophic vs oligotrophic lakes. Perhaps it would be easier for readers if the authors provide a table of the individual lakes as a supplemental material.

The discussion needs major reorganization. It is very easy for readers to get lost in the message that the authors are trying to convey.

Additional comments

more direct and active, then readers would still feel engaged.
Lines 24-28: This is a little hard to read and readers will get lost. Perhaps the authors would consider shortening this so readers can quickly understand the type of results to be expected and what they imply?
For example:
This study couples limnological characterization with high-throughput 16S rRNA sequencing of (V4 hypervariable region) to assess microbial composition and diversity in oligotrophic versus eutrophic lakes.
Lines 29-30: Title suggests diversity changes between trophic conditions.
Line 67-70: Why are study this set of lakes? The importance of these lakes is muddled by the description. It would help readers understand the importance of why these lakes need to be studied if lines 76-81 were integrated into this paragraph -- these are recreational lakes and eutrophication in these lakes pose a threat to visitors. Furthermore, it is a protected natural area, which signifies the importance to a community that relies on it as a source of freshwater.
Line 73-75: What is the goal of this sentence? Transition to this sentence is abrupt. Are the authors trying to convey that agriculture is important yet it has a negative impact on the lakes?
Line 97: Previously defined as “chl-a” in line 27. Please go through the manuscript and make sure this is consistent. There are several “chlorophyll-a” and “chl-a” throughout the manuscript.

Line 97: Caution should be used when addressing productivity. Primary productivity is often represented as how much photosynthesis is conducted by phytoplankton. More chlorophyll does not mean more primary productivity because there are other factors associated with primary productivity. Furthermore, phytoplankton can undergo photoinhibition.

Gilbert, M., Domin, A., Becker, A. et al. Estimation of Primary Productivity by Chlorophyll a in vivo Fluorescence in Freshwater Phytoplankton. Photosynthetica 38, 111–126 (2000). https://doi.org/10.1023/A:1026708327185

Line 150: Correction for phaeopigment? Not all researchers will correct for this so please make sure your readers know what steps were taken so they can follow your steps.
Line 151: I’m not sure why there are two citations of the same thing. EPA method 445.0 the same as Arar and Collins 1997. It is very rare to cite an EPA method and papers usually site the author

Line 156-157: Do the authors mean to say that PCA was first used to identify important parameters to be further used in the CA? If so, please clarify this because it may be confusing to readers who are not familiar with environmental statistics. Current statements sounds like the CA was used first then the output from the CA was used in the PCA to identify the most important parameters.

Line 191: The authors needs to clarify that it is SILVA version 132 SILVA release 132) and not use the “-“ after because it will confuse readers since there is another value after the “-“

Line 193-202: There are many ordinations available from Phyloseq (nmds, CA, CCA etc). Were all methods used? It is hard to follow what is being used because CAP is a method that involves two types of ordinations. The authors need to be clear which package was used for these ordinations. Both Phyloseq and vegan packages have the capability to do ordinations but Phyloseq is less user friendly whereas vegan requires reformatting the dataframe such that the ordinations can be plotted with ggplot. Furthermore, vegan has more ordination options than Phyloseq.

Line 199-202: Isn’t CAP an statistical method that uses two ordinations? If so, then would it make more sense to put this in the statistical analysis? It is easy for readers to get lost because there is a statistical analysis section, then a bioinformatics section and a return to some statistics analysis.

Line 202-204: Was the PERMANOVA used on the sequence dataset or the physiochemical dataset? Based on the results, it seems to be used to assess if differences in physiochemical parameters. Is this method used for oligotrophic vs eutrophic lakes or for each lake? “…different lakes” needs to be clarified.

Line 219-223: This paragraph is random and the transition is very abrupt.

Line 244-249: A PCA was never mentioned in the methods. I am curious what the percentages are for PC1 vs PC2? If low K values lie on the right side of PC1, then why does line 51-52 claim that lakes with high K lie to the right of PC1? If K is a primary contributor to PC1 and the right side is high K, then I suspect it might that PC2 is more influence on group B than PC1. If the PCA was done in R, then the authors can check the contributions of individual samples.

Also, PCA assumes normal distribution. I am assuming that the distribution improved after normalizing values to the z-score?

Line 246: “however” is used to show contrast and may not be appropriate here. PC1 is primarily contributed by K and PC2 are primarily contributed by Chl-a and Zeu. PC1 and PC2 are not contrasting each other and neither are the variables associated with each PC.

Line 287: This needs clarification. Is the microbial composition widespread throughout the water column?

Line 292: Trophic conditions are based on the TSI index (Secchi Depth, Chl-a, and phosphorus). Are the lakes classified based on microbial composition or trophic status? This part needs to be clarified.

Line 306: Which analysis?

Line 308: Earlier names in this sentence have “sp.” but not Cyanobium? Are the authors saying certain Planktothrix and Acinetobacter species all Cyanobium species?

Line 334-337: According to the data (taken in winter), not all lakes are mixed in the winter season? Some lakes have anoxic bottom waters.

Lines 346-347: Nutrients are tricky. The N:P ratio plays a large role in phytoplankton growth. If there is excess N, then there needs to be sufficient P for phytoplankton to utilize the N. If there is not enough P, then phytoplankton will not grow even if N is in excess.

Lines 359-363: Yes, chl-a can be monitored using remote sensing (RS) and in situ probes but I’m not sure how that is relevant to this study. There are limitations of using RS to monitor chl-a.

Line 386: Representative of oligotrophic lakes?

Line 387-388: This needs to be clarified to readers. Readers can interpret this sentence as the two lakes are eutrophic lakes.

Lines 378-393: The authors need to clarify their message in this section. There are multiple ideas going one and readers can get lost in the message. The Dos Lagos and Esmeralda lakes have interesting findings that warrant further research since the PCA grouped these with eutrophic lakes. Their microbial compositions are representative of oligotrophic conditions, yet chl-a concentrations are in the middle of oligotrophic and eutrophic status.

Lines 402:421: It is easy to get lost in this section. Previous paragraph is a transition paragraph that pivots the focus to cyanos and this section opens up with cyanos and eutrophy. Then the section moves onto Proteobacteria and oligotrophic conditions. It is hard to identify the topic sentence in this section because there’s too many ideas going on.

Figure 1: Perhaps the authors would consider an active caption: what does the comparison tell the reader? For someone who is colorblind, it may not be so obvious that there is an increase in agricultural landuse since 1992. By using an active caption, readers can better grasp the message the authors are trying to convey.

Figure 2: Red dots in green background may not be easily identified by readers who are colorblind. Either change the colors to create a set of symbols for eutrophic/oligotrophic lakes

Table 1: Chl-a column is cut off

---

## Round 0.2 · Minor Revisions

Reviewer 2 has provided extensive suggestions to improve the manuscript, and has offered generously to review a final draft. I support all of their conclusions and suggestions and encourage you to provide an extensive rebuttal document addressing R2's suggestions point by point.

Reviewer 1 ·

Basic reporting

The authors have addressed all my previous comments and the manuscript has been substantially improved. I have no further comments to add.

Experimental design

no comment

Validity of the findings

no comment

Reviewer 2 ·

Basic reporting

In this revised manuscript, the authors compared physicochemical parameters and microbial community composition in oligotrophic versus eutrophic lakes. Based on the revision, the introduction has improved tremendously and the figures as well. However, the reference list needs to be thoroughly checked. Some references are not mentioned in the manuscript and it probably was in the original manuscript but removed and the reference list was not updated. There are some minor grammar suggestions that could improve the manuscript (in comments below). Overall, the authors have made efforts to improve this manuscript.

Experimental design

Based on the revision, the research question and goal are clear now. Their hypothesis is clearly explained. The addition of background information regarding the geological region surround the lakes is sufficient for readers to understand the authors' perspective. The separation of a sub-section for statistical analysis makes it clear for readers to understand their steps in analyzing the data.

Validity of the findings

Based on the revision, the manuscript has improved. However, there are some minor points (easily fixable) that needs to be addressed. Every paper requires a good flow so the reader can follow the logic in the manuscript. The first sub-section of the discussion needs some rearranging so readers can easily follow the manuscript. The authors found a correlation between chlorophyll and trophic status but need to clearly relate this to monitoring (similar is applied to water clarity parameters). In the second sub-section of the discussion, it is unclear what the relationship is between chlorophyll and microbial community and the authors need to clarify this.

Additional comments

Dear Authors,
I am pleased to review this MS again. I believe the topic is worthwhile and would be valuable in the progression of science. To get the MS ready for readers, I have made the following suggestions. Please note that most of the recommendations are to improve the MS and not be a critical statement. I believe that reviewers can offer a set-of-eyes that are closer to the reader than the authors. As such, I offer these suggestions:

1. Discussion – Physicochemical characterization of the lakes of the LMNP
Again, every paper requires a good flow so the reader stays engaged.
Based on the title and line 334-336, readers are expecting a discussion about the physiochemical characteristics of the lakes - which the authors do a great discussion about the parameters. However, there are random mentions of monitoring - lines 352,363-367, and 378 also hints this and these stand out like a sore thumb.
Perhaps the authors are trying to say that the physiochemical parameters suggest that these parameters need long term monitoring? If so, then this entire section would flow better if the authors start with eutrophication, discuss the parameters - DO, Chl-a, water clarity, then return to eutrophication and finally end with "long-term efforts to understand the seasonal changes and magnitude in eutrophication of these lakes is needed" and monitoring

It would be fairly easy to reorganize the thoughts/lines closer to a "funnel" this section of the discussion - from largest to more focused subjects. i.e., Eutrophication are measured by xzy. Evidence of eutrophication – discuss the parameters. Eutrophication was first observed in 2003. Then end with the need to monitor these parameters/long term efforts.
Following this prescribed structure will be reasonable to the reader, and as such, the reader will follow the logic of the MS better.

2. Please check that all crossed reference figures are correct and all figures/tables are properly labelled. E.g., Word doc Table S4 and PDF Table S5 are the same.

3. Please check references list. There are some references that have not been cited in the MS at all but somehow ended up in the references.

4. I would caution making general statements. Line 402 states that Planktothrix is usually dominant in eutrophic lakes. However, this can be easily refuted with a quick literature search. The dominant cyanobacteria can shift to a different genus as nutrients/physical conditions change over time. Furthermore, not all eutrophic lakes are dominated by Planktothrix – e.g., microcystis, dolichospermum. Yes, Davis et al found Planktothrix as the dominant genus in a small area of Lake Erie but that group also noted Microcystis in the southern part of the lake.

Jankowiak, Jennifer, Theresa Hattenrath‐Lehmann, Benjamin J. Kramer, Megan Ladds, and Christopher J. Gobler. 2019. “Deciphering the Effects of Nitrogen, Phosphorus, and Temperature on Cyanobacterial Bloom Intensification, Diversity, and Toxicity in Western Lake Erie.” Limnology and Oceanography 64 (3): 1347–70. https://doi.org/10.1002/lno.11120.

The following are comments in Response to Reviewers.
1. Certainly, the MS is more complete and has a more direct message. Their reworking the MS to include details in the results and discussion gave scientific vigor to the MS.
2. The Introduction is now more complete and focused. Rather than being unclear in their goal and research question, readers can easily follow their message. I appreciate the authors’ efforts to add more background information regards to the study sites. There are many types of lakes, located in various geological regions. We, as readers, interpret the message based on our own experience, therefore the background information is vital to understanding the authors’ perspective.
3. The new discussion still needs some revision, although, small. Our choice of words strongly influence how readers can interpret any researchers’ message.
4. Overall, the edits have removed most of my concerns about this manuscript. I would like to see this MS published.

Specific comments:
Line 21: Grammar: Signs of eutrophication
Line 22-23: Grammar: Anthropogenic activities alter the water quality and trophic states of lakes in LMNP
Line 149: This has two references. See Author guidelines:
https://peerj.com/about/author-instructions/#reference-format
Line 182: Needs space after "of"
Line 209-210: ..had statistical differences ….
Line 262: These are the same table. Table S4 is a word document that shows the same content as the PDF Table S5. Now the word doc "Supporting Information" contains Table S4, which shows alpha diversity but there is no Table S5. Please check the cross-references and Table numbers
Lines 336-341: Choice of words can me a huge difference in easily the reader can follow the authors' message. Readers may have a hard time understand the flow in Lines 336-341:
Line 336-337: "when the lakes should have experience full mixing" can give the impression that it did not fully mixed otherwise
Line 338: "no density barriers...nonetheless, mixing requires ...wind…" To a reader, this can be interpreted as density barriers are absent but it will not mix because mixing requires wind. "Nonetheless" is used to show contrast, similar to "however"
Line 339-341: The DO profiles show mixing
Line 340: Grammar: Then water column….
Line 335-336: "In contrast to oligotrophic lakes" or "Different from oligotrophic lakes"
Lines 360-366: Again, I am not sure what the goal of this paragraph is. It is unclear what message the authors are trying to convey. The first sentence talks about water clarity/transparency between eutrophic/oligotrophic lakes then the authors suggest using data from monitoring water clarity/transparency to quantitatively compare both chl-a and water clarity between lake types. Are the authors saying that they can't compare water clarity between types of lakes?
Line 389: Again, check your tables, PDF is different than the supporting info word doc
Line 390-397: What is the relationship between chl-a and microbial diversity/composition? This needs to be clarified.
Line 391: Grammar: with regards to eutrophic lakes
Fig 6 caption: Wrong symbol in figure. Check symbols in text match those in the figure.
Fig. 2: Grammar: Simbology, should be Symbology or Legend
Fig 6: Lake names are not consistent with other figures (i.e., other figures do not have underscore "_")

---

## Round 0.3 · accepted · Accept

Thank you for addressing all of the reviewers' concerns and revising your discussion to clarify the relationships between chlorophyll and microbial communities.